# Inhibition of the Growth of *Botrytis cinerea* by *Penicillium chrysogenum* VKM F-4876D Combined with Fludioxonil-, Difenoconazole-, or Tebuconazole-Based Fungicides

Amjad Hatem, Vera Yaderets, Nataliya Karpova 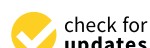, Elena Glagoleva, Alexander Ovchinnikov, Kseniya Petrova, Alexandra Shibaeva and Vakhtang Dzhavakhiya *

Federal Research Center "Fundamentals of Biotechnology", Russian Academy of Sciences, 117312 Moscow, Russia; amjadhatem82@gmail.com (A.H.); verayaderetz@yandex.ru (V.Y.); ashatanr@mail.ru (N.K.); glagolevaev@mail.ru (E.G.); centr-biotech@mail.ru (A.O.); petrova.ksenia.s@yandex.ru (K.P.); aleksandrashibaeva@mail.ru (A.S.)
* Correspondence: vahoru@mail.ru

**Abstract:** *Botrytis cinerea,* causing grey mold, is a dangerous plant pathogen able to infect agricultural crops during the whole production cycle, including storage and transportation. A wide set of pathogenicity factors, high ecological plasticity, and universality of propagation and spreading of this fungus significantly complicate the control of this pathogen. A rapid increase in pathogen tolerance to fungicides dictates the necessity of developing antiresistant protection strategies, which include the use of biopreparations based on antagonistic microorganisms or their metabolites. The purpose of the study was to evaluate the antifungal activity of a dry biomass of *P. chrysogenum* VKM F-4876D (DMP), both individually and in combination with tebuconazole-, fludioxonil-, or difenoconazole-containing compounds recommended to control grey mold, in relation to *B. cinerea* isolated from grape samples. A water suspension of DMP was added to the PDA medium at a concentration of 1.0, 2.5, 5.0, 7.5, and 10.0 g/L. The pathogen growth inhibition was evaluated after 3, 7, and 14 days of cultivation; fungal cultures grown on DMP-free medium were used as a control. The resulting effective DMP concentration was 2.5 g/L. The effective concentrations of fungicides included in the study were determined to be 0.5 mg/L (tebuconazole), 0.1 g/L (difenoconazole), and 0.04 mg/L (fludioxonil). Combining DMP (2.5 g/L) with tebuconazole, difenoconazole, or fludioxonil (all taken at the effective concentrations) resulted in pathogen growth inhibition after 7 days of incubation by 86.5, 85.6, and 84.6%, respectively. Among all studied variants, the DMP (2.5 g/L) + difenoconazole (1.0 mg/L) combination provided the most efficient control of *B. cinerea* development under in vitro conditions: even after 14 days of incubation, the pathogen growth suppression remained at the level of 51.3%, whereas the DMP combination with tebuconazole or fludioxonil provided only 28.5 and 37.4%, respectively. The obtained results show good prospects for the efficient control of grey mold development, together with the reduction of pesticide loads on agrobiocenoses and the prevention of the emergence of new resistant forms of plant pathogens.

**Keywords:** *Botrytis cinerea*; biopreparations; biocontrol methods; plant protection

## 1. Introduction

*Botrytis cinerea* Pers. Fr is a universal necrotrophic plant pathogen belonging to the genus *Botrytis* of the family *Sclerotiniaceae*. This pathogen has a wide range of host plants and infects flowers, leaves, fruits, shoots, tubers, and other plant organs of more than 1400 species of ornamental and agricultural crops from 586 genera [1–5]. In this regard, *B. cinerea* is similar to the phylogenetically released *Sclerotinia sclerotiorum*, which causes white mold in various plant species [5,6]. Though *B. cinerea* is usually considered to be less aggressive than *S. sclerotiorum*, its ability to spread rapidly and its high adaptability make it an economically important pathogen (high-risk pathogen) able to cause significant

yield losses at both vegetation and low-temperature (0–10 °C) storage periods [1,5,7]. *B. cinerea* causes a disease called "grey mold" [7,8]; it infects mainly stored vegetables, grapes, and berries [9,10]. High humidity and a low temperature facilitate rapid spreading of the pathogen.

The control of a *B. cinerea* distribution is complicated because of its ability to form sclerotia with a high content of melanin and β-glucan, protecting them from drying, UV radiation, and other negative environmental factors and initiating new infection cycles [3,9,11,12].

One of the best ways to suppress grey mold on agricultural crops is through the application of fungicides differing in their structure and mechanisms of action [13–15]. However, a wide use of chemical fungicides caused a reduction in their efficiency because of the development of resistant *B. cinerea* strains [14,15]. The phenomenon of the development of *B. cinerea* resistance to fungicides is the object of thorough studies in different countries [16]. For example, 474 commercial fungicides and their combinations intended to control grey mold were registered in China in 2020; these fungicides belonged to different classes, including quinone outside inhibitors (QoI), anilinopyrimidines (AP), methyl benzimidazole carbamates (MBC), dicarboximides (DCF), phenylpyrroles (PP), and succinate dehydrogenase inhibitors (SDHI). The fungicide resistance monitoring performed in 2003–2017 showed that isolated *B. cinerea* strains demonstrated various levels and frequencies of resistance to common fungicides. In some cases, MBC and DCF fungicides could not be further used to control grey mold [17]. Note also that the treatment of stored vegetables, fruits, and berries with chemical fungicides is not recommended [18]. Due to these reasons, agricultural companies often use environmentally safe methods and preparations, considered alternatives to chemical protection, to reduce the volume of post-harvest yield losses and to provide prolonged storage of fresh products [19,20]. Moreover, the accumulation of residual quantities of fungicides in the soil, water, and agricultural production poses a significant threat for humans, animals, and the environment [4,21,22].

Within the framework of the existing restrictions on the use of synthetic pesticides, the development of methods supplementing chemical control and based on the use of nonpathogenic microorganisms or their metabolites is often considered a promising alternative. Such preparations are the most suitable for inclusion in integrated plant protection because they are quite effective, selective, and safe [9,23]. Among microorganisms used for this purpose, those from the genera *Penicillium* and *Bacillus* are especially interesting since they synthesize various biologically active compounds [24,25] and are characterized by a wide range of antifungal activity [26,27]. Moreover, preparations based on these microorganisms are capable of growth-stimulating action (to some extent) and inducing plant defense mechanisms. For example, biopreparations for grape protection based on *Bacillus subtilis*, *Aureobasidium pullulans*, and *Bacillus amyloliquefaciens* have been registered in France [16]. However, according to a number of studies, the field efficiency of these biopreparations and the dependence of this efficiency on the season as well as various climatic and/or local agronomic conditions are still in question. Similarly, some published research papers report insufficient efficiency of some of such products in viticulture, even in the case of multiple treatments [16].

Thus, in spite of the practice of introducing bio-based preparations for plant protection, a complete refusal to use modern fungicides is rather impossible and commercially unjustified since they are able to provide efficient control over crop diseases [28]. In this regard, the use of fungicides combined with biologically active compounds of microbial or plant origin (as well as their synthetic analogues) may represent a rather promising approach for effective crop protection [29,30]. A synergistic or additive effect provided by the interaction of such preparations may result in a significant reduction of the working fungicide concentrations to levels, at which they are inefficient if applied alone, without any losses in their efficiency [29].

Earlier, the authors of this study developed a *P. chrysogenum* strain, F-4876D, and showed that the use of its dry mycelial biomass (DMP) was able to suppress the growth and development of some plant pathogenic fungi [31]. The purpose of this study was the

evaluation of the ability of a dry mycelial biomass (DMP) of a *P. chrysogenum* VKM F-4876D strain, taken alone or combined with difenoconazole-, tebuconazole-, and fludioxonil-based fungicides, to suppress the growth and development of a *B. cinerea* strain, isolated from fresh grape berries.

## 2. Materials and Methods

### 2.1. Reagents

Commercial nutrient media used for the cultivation of fungal strains included potato dextrose agar (PDA; Condalab, Madrid, Spain) and Czapek Dox Agar (HiMedia Laboratories, Mumbai, India).

Fungicides included in the study were Folicur® EC 250 (a.i. tebuconazole, 250 g/L; Bayer AG, Leverkusen, Germany), Maxim® SC (a.i. fludioxonil, 25 g/L; Syngenta, Saint-Pierre la Garenne, France), and Skor® EC (a.i. difenoconazole, 250 g/L; Syngenta, Basel, Switzerland).

Inorganic salts, glycerin, and glucose used in the study were purchased from Acros Organics (Geel, Belgium). Agar was manufactured by Difco (Detroit, MI, USA). Skimmed deodorized soybean flour (Soyanta-200) was purchased from Yanta (Irkutsk, Russia). Meat peptone was purchased from HiMedia Laboratories (Mumbai, India).

### 2.2. Microorganisms

*P. chrysogenum* VKM F-4876D strain was provided by the work collection of the Laboratory of Biotechnology of Physiologically Active Compounds of the Federal Research Center "Fundamentals of Biotechnology" (Moscow, Russia). A *B. cinerea* strain was isolated from grape berry samples (cv. Cardinal) collected on the experimental plots of the All-Russian National Research Institute of Viticulture and Winemaking "Magarach" (Yalta, Republic of Crimea, Russia).

### 2.3. Isolation and Identification of Botrytis cinerea

*B. cinerea* was isolated from grape berries with typical grey mold manifestations. Berry samples were put into sterile containers and stored for 10 days at 4 °C. Then the plant material was transferred onto Petri plates with PDA medium and incubated for 3–5 days at 24 °C in the dark. Small (5 × 5 mm) agar blocks were cut from the grown mycelium, transferred to the center of Petri plates containing fresh PDA, and cultivated as described above. The resulted fungal colonies were used to derive monoconidial isolates, which were grown for 5–7 days under similar incubation conditions with alternating light/dark periods.

Morphological identification of *B. cinerea* was performed via microscoping of 3–7-day colonies at 100× magnification to determine the presence/absence of spores, branching type, and form of conidia [32,33]. Molecular identification of *B. cinerea* was performed at the Molecular Diagnostics Laboratory of the Federal Research Center "Fundamentals of Biotechnology" of the Russian Academy of Sciences (Moscow, Russia).

DNA isolation from biomass samples was carried out according to the earlier described method [34], with the addition of zymolyase (20–50 U per sample). The concentration of the resulting DNA preparation was 30–50 μg/mL, while RNA was presented only in trace amounts (<1%).

The ITS5-LR5 primer system, which provided the amplification of a DNA region including the intergenic region, the 5.8S RNA gene, and a fragment of the 28S RNA gene, was used to perform PCR. Isolation and purification of PCR products from 2% agarose were carried out using a Wizard PCR Preps DNA purification kit (Promega, Madison, WI, USA) according to the manufacturer's recommendations.

The sequencing of the obtained PCR fragments was carried out according to Sanger et al. [35] using a BigDye terminator v.3.1 kit (Applied Biosystems Inc., Foster City, CA, USA) and an Applied Biosystems 3730 DNA analyzer (Applied Biosystems Inc., Foster City, CA, USA).

A phylogenetic analysis of the obtained nucleotide sequences was performed using the BLAST algorithm of the NCBI GenBank database (http://www.ncbi.nlm.nih.gov (accessed on 5 October 2023)). The sequence alignment and editing were carried out using the

Bioedit program package (http://jwbrown.mbio.ncsu.edu/BioEdit/bioedit.html (accessed on 5 October 2023)).

*2.4. Composition of Nutrient Media for Cultivation and Maintenance of Penicillium chrysogenum and Botrytis cinerea*

A *P. chrysogenum* VKM F-4876D strain was maintained on the following medium (g/L): agar, 20.0; glucose, 30.0; glycerol, 70.0; soybean flour, 10.0; meat peptone, 10.0; NaNO$_3$, 2.0; and MgSO$_4$·7H$_2$O, 1.0 (pH 6.3–6.5).

A *B. cinerea* isolate was maintained on a ready PDA medium for one month at 4 °C, then re-inoculated onto fresh medium. For long-term storage, the fungus was put into 50% glycerol and stored at −20 °C.

For the spore germination experiment, a potato-glucose broth was used. Peeled potato (200 g) was boiled for 1 h in 1 L of water. Then the broth was passed through a cotton-gauze filter and supplemented with glucose (20 g) and water up to a final volume of 1 L (pH 6.0). The medium was poured into 750-mL round-bottom flasks (250 mL per a flask). The flasks were plugged with cotton plugs and sterilized for 20 min at 121 °C and 15 psi.

*2.5. Cultivation of Penicillium chrysogenum F-4876D in Liquid Medium and Obtaining of Dry Mycelial Biomass (DMP)*

Ten mL of a sterile physiological solution was poured into a tube containing a F-4876D culture. The top mycelial layer was accurately taken by a microbiological loop and transferred into 750-mL flasks with 100 mL of the following medium (g/L): sucrose, 100.0; soybean flour, 20.0; trypton, 10.0; NaNO$_3$, 2.0; MgSO$_4$ · 7H$_2$O, 1.0; distilled water, 1 L (pre-sterilization pH 5.7–6.0). The flasks were incubated for 48 h on an Innova 44 incubation shaker (New Brunswick, Germany) at 24 °C and 220 rpm.

After completion of incubation, 10% of the obtained inoculate was added into 1-L flasks containing 250 mL of the following medium (g/L): sucrose, 100.0; soybean flour, 30.0; trypton, 10.0; NaNO$_3$, 2.0; MgSO$_4$ · 7H$_2$O, 1.0; distilled water, 1 L. The flasks were placed on the same shaker and incubated for 4 days under the same conditions. Then the resulting *P. chrysogenum* biomass was inactivated for 30 min at 80 °C and freeze-dried.

*2.6. Evaluation of the Antagonistic Activity by a Dual Culture Assay*

The antagonistic activity of *P. chrysogenum* F-4876D towards *B. cinerea* was determined by a dual culture assay. Spore suspension of each microorganism ($1 \times 10^6$ spores, 0.1 mL) was added onto a Petri plate and spread over the PDA surface with a sterile spatula. Plates, inoculated with the F-4876D strain and the pathogen, were incubated at 24 °C for 5 and 3 days, respectively. Then agar blocks (1 cm in diameter) containing fungal mycelium were cut under sterile conditions from plates inoculated with *P. chrysogenum* F-4876D and transferred onto the surface of fresh plates with PDA. Since this fungus grows on PDA rather slowly, inoculated Petri plates were returned to a thermostat for 3 days. Then agar blocks containing *B. cinerea* mycelium were placed at a distance of 5–6 cm from *P. chrysogenum* blocks, and plates were further incubated under the same conditions. Petri plates containing agar blocks with *P. chrysogenum* or *B. cinerea* only were used as the controls.

The antagonistic evaluation was carried out after 7 days of joint cultivation. The experiment was arranged in three replications, each including three Petri plates per variant. All calculations were performed using arithmetic means.

The growth inhibition level (*GI*) was quantified using the following formula:

$$GI = \frac{d1}{d2} \cdot 100\% \tag{1}$$

where *d*1 is the distance between the center of the agar disc with *B. cinerea* and the edge of its colony growth, and *d*2 is the distance between the centers of agar blocks containing *P. chrysogenum* F-4876D and *B. cinerea*.

### 2.7. Preparation of Agar Medium Supplemented with DMP Suspension

Five grams of DMP were supplemented with 50 mL of sterile physiological solution and mixed for 1 h at 28 °C and 220 rpm. Then the obtained suspension was added to sterilized PDA medium by a sterile pipette to obtain the final concentration in accordance with the experimental scheme. After thorough mixing, the medium was poured into Petri plates.

### 2.8. Preparation of Agar Medium Supplemented with Fungicides

Aliquots (100 μL) of each of the fungicides included in the study (Folicur® EC (Bayer AG, Leverkusen, Germany), Maxim® SC (Syngenta, Saint-Pierre-la-Garenne, France), Skor® EC (Syngenta, Basel, Switzerland)) were taken from commercial fungicide samples, placed into 1.50-mL Eppendorf tubes containing 900 μL of sterile distilled water, and thoroughly mixed. After a series of sequential dilutions, the final fungicide solutions (0.01, 0.1, 1.0, and 10.0 mg/mL) were obtained. After sterilization, agarized medium was supplemented with 100 μL of the corresponding fungicide solution up to the final concentration (according to the experimental scheme), mixed, and poured into Petri plates.

### 2.9. Evaluation of the Antifungal Activity of DMP by a Radial Growth Method

The radial growth evaluation assay was performed as described in [31]. To obtain agar disks, 0.1 mL of the spore suspension of *B. cinerea* ($1 \times 10^6$ cells/mL) was dropped onto the surface of a sterile PDA medium and spread by a sterile spatula. After a 24-h incubation of inoculated plates at 24 °C, 10-mm disks were cut from the medium using a sterile borer and placed at the center of fresh Petri plates (d = 85 mm) with PDA medium supplemented with DMP or a combination of DMP and a fungicide. Plates with the PDA medium were used as the control. Inoculated plates were incubated at 24 °C for 14 days. The measurement of average colony diameters in the control and experimental variants was carried out on days 3, 7, and 14 of incubation. The average colony diameter was determined as the arithmetic mean of the maximal and minimal colony diameters.

The experiment was arranged in three replications, each including three Petri plates per variant. All calculations were performed using arithmetic means.

The antifungal activity (*AFA*) was determined by the following formula:

$$AFA = \left( 1 - \frac{D_E}{D_C} \right) \cdot 100\% \tag{2}$$

where $D_E$ and $D_C$ are colony diameters (mm) in the experimental and control variants, respectively.

Petri plates inoculated with the pathogen were considered the controls.

A synergistic effect was calculated using Limpel's criterion [36] according to the following formula:

$$E_E = X + Y - \frac{XY}{100} < E_R \tag{3}$$

where $E_E$ is the expected additive effect of use of both compounds (%), $E_R$ is the effect obtained in the experiment for a joint use of both compounds (%), and *X* and *Y* are growth-inhibiting activities provided by each of the tested compounds used alone (%).

The effective concentration was determined graphically on the seventh day of the experiment using probit analysis [37] and Microsoft Excel 2019 program package.

### 2.10. Evaluation of the DMP Effect on the Conidial Germination of Botrytis cinerea

The DMP effect on conidial germination was evaluated in flat-bottomed 96-well plates (SPL Life Sciences, Pocheon-si, Republic of Korea). Ten milliliters of potato-glucose broth were added into a tube with a 7-day *B. cinerea* culture grown on PDA, and the top mycelial layer was scraped by an inoculation loop to obtain the pathogen's conidia. To remove mycelium fragments, the resulting suspension was filtered through sterile cotton wool. To determine the concentration of conidia, a drop of suspension was placed onto a Goryaev

chamber and microscoped at 100× magnification using a Primo Star microscope (Carl Zeiss, Oberkochen, Germany). For further work, the conidial suspension was adjusted (if necessary) to a concentration of $5 \times 10^4$ conidia/mL using the same nutrient medium.

To determine the concentration dependence of the DMP effect on the germination of conidia, a series of double dilutions was prepared within the DMP concentration range of 0.625–10.0 mg/mL. DMP-free potato-glucose broth was used for the control. Each well was filled with 100 μL of the corresponding DMP solution and an equal volume of conidial suspension, so the final conidial concentration in each well reached $2.5 \times 10^4$ conidia/mL. The plate was placed into a thermostat and incubated for 24 h at 25 °C, then cells were fixed with 50 μL of 2% paraformaldehyde solution and microscoped at 100× magnification. The experiment was organized into three replicates.

### 2.11. Inhibition of Botrytis cinerea Development on Injured Grape Berries in the Presence of DMP

The effect of DMP on the development of *B. cinerea* on grape berries was studied as described in [38]. Mature grape (cv. Husain, Uzbekistan) was purchased in a local shop in Moscow (Russia). Calibrated berries (average length of 19–20 mm) were preliminarily soaked in a 1% NaOCl solution for 2–3 min, then twice washed with sterile distilled water and dried on filter paper under sterile conditions. Each berry was injured by a sterile wooden stick (2 mm in diameter). All berries were divided into control and experimental groups, each consisting of 10 berries. Berries from the experimental and control groups were soaked in a DMP suspension in sterile water (2.5 g/L) or in sterile water, respectively. After 10 min, all berries were placed on filter paper under sterile conditions until complete drying of their surface, and 20 μL of *B. cinerea* spore suspension ($10^4$ spore/mL) were inoculated into each injury. The spore suspension was obtained as described in 2.7, except for the use of sterile water instead of potato broth. Inoculated berries were put into sterile plastic containers and incubated in the dark at 24 °C for 12 days. The results of the experiment were estimated on days 3, 7, and 12. The experiment was arranged in three replications, each including two series of the control and experimental variants. During observations, the injury level of each berry was evaluated in accordance with a 10-score visual estimation scale (Figure 1). The disease incidence (*DI*) was quantified using the following formula:

$$DI = \frac{n}{N} \cdot 100\% \tag{4}$$

where *n* is the number of berries corresponding to a certain injury level and *N* is the total number of berries.

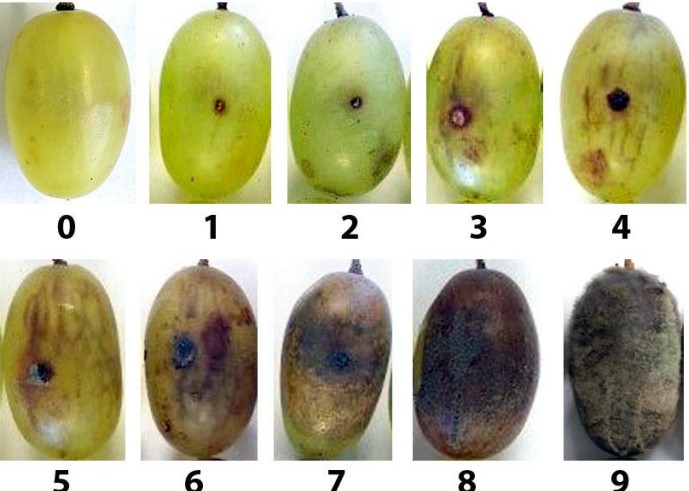

**Figure 1.** Visual scale for estimation of a grape berry infection with *B. cinerea* (0–9-scale, where 0 corresponds to a healthy berry).

### *2.12. Data Treatment*

All results shown in diagrams represent arithmetical means and standard errors. To determine the tightness and statistical significance between the amount of added fungal biomass and the antifungal effect, a Pearson correlation coefficient was calculated at the *t*-test significance level of 0.01. The obtained data were statistically treated using the MS Excel 2019 software package. The significance of the difference between different samplings was determined by the least significant difference method.

## 3. Results

### *3.1. Identification of Isolated Botrytis cinerea*

Identification of the isolated pathogen was performed using morphological criteria proposed by other authors [32,33], with allowance for features of conidiophore branching and the form and size of conidia.

Colonies grown on PDA were characterized by rapid growth, were dark-grey and felty, and formed regular circles whose diameter reached 4–4.5 cm on the fourth day of cultivation. No sclerotia were observed. The hyphae were thin and septated. Conidiophores were well distinguished from vegetative hyphae; they were tree-shaped with short branches carrying smoke-colored conidia (9.0–17.50 × 6.50–10.0 μm in size) collected into dense clusters or heads. Conidia were oval-shaped with a thin beak near the base (Figure 2). The optimal temperature for colony growth was 22–24 °C, with complete growth inhibition at 28 °C.

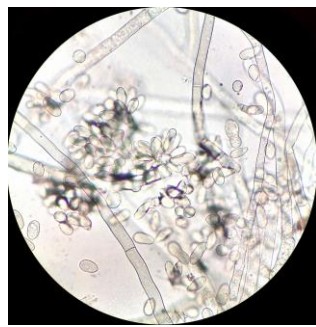 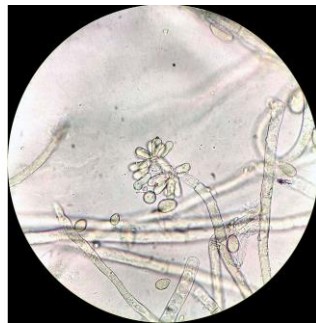

**Figure 2.** Microscopic image of *Botrytis cinerea* isolated from grape berries (100× magnification).

The molecular identification of the strain was carried out using the ITS5-LR5 primer system, allowing amplification of a DNA region containing the intergenic region as well as genes encoding 5.8S rRNA and (partially) 28S rRNA (Figure S1, Supplementary Materials). The performed BLAST analysis showed that the obtained 1388-bp sequence of the studied strain was the most similar to those of *B. cinerea* strain SI3 (99.93%) and *B. cinerea* strain B05 (99.86%). The search algorithm covered nucleotide sequences from the NCBI GenBank database (http://www.ncbi.nlm.nih.gov (accessed on 5 October 2023) available as of 10 July 2023 (except for non-cultivated pathogens)).

Based on the morphological traits of the isolate and the results of its molecular identification, we concluded it belonged to the *B. cinerea* species.

### *3.2. Evaluation of the Antagonistic Activity of F-4876D towards Botrytis cinerea*

In spite of the weak growth of *P. chrysogenum* on PDA medium, its metabolites released into the agar medium inhibited the colony growth of *B. cinerea* (Figure 3).

The antagonistic antifungal activity calculated for the seventh day of co-cultivation using Formula (1) (see Section 2.5) was 55.6%. The microscopic study of the area of *B. cinerea* contact with the inhibition zone showed that, due to *P. chrysogenum* metabolites released into the agar medium, *B. cinerea* hyphae are deformed, thickened, and more septated as compared to the control (Figure 3D,E). Almost a complete absence of conidia was observed.

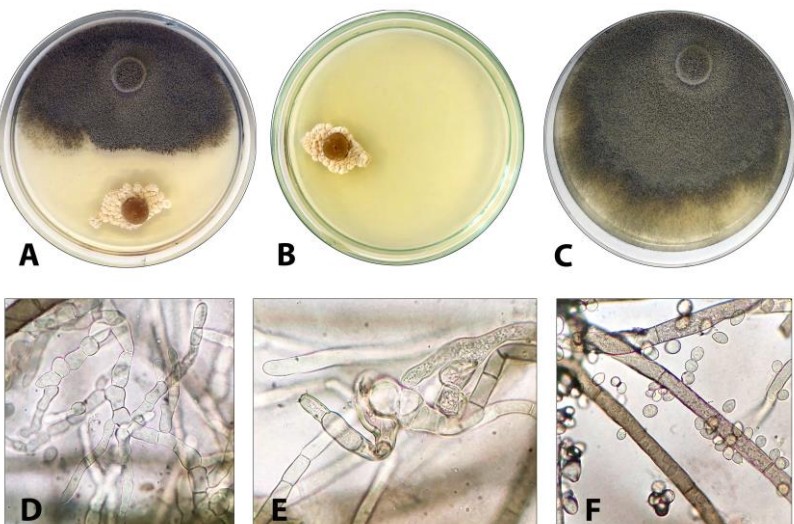

**Figure 3.** (**A**) Antagonistic activity of *Penicillium chrysogenum* F-4876D in relation to *Botrytis cinerea* after a 7-day co-cultivation; (**B**) growth of *P. chrysogenum*; (**C**) growth of *B. cinerea*; (**D,E**) hyphae of *B. cinerea* contacting the inhibition zone; (**F**) control (100× magnification for (**D–F**).

### 3.3. Antifungal Activity of DMP towards Botrytis cinerea

A possible dependence between the DMP concentration in the nutrient medium and the level of growth inhibition of *B. cinerea* was determined by a radial growth method. The effect was evaluated at the end of a 7-day cultivation, when pathogen colonies spread up to the edges of Petri plates in the control. The results of the experiment calculated using Formula (2) are shown in Figure 4.

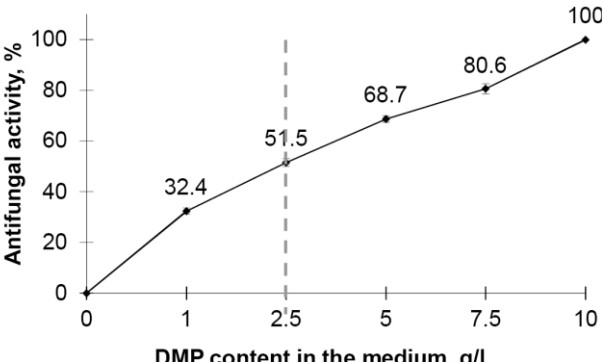

**Figure 4.** Dependence of the growth of *Botrytis cinerea* colonies on the DMP concentration in PDA medium. The area located to the right of the dotted line indicates the zone of effective DMP concentrations with antifungal activity equal to or exceeding 50%.

According to the obtained results, the presence of DMP within the concentration range of 5.0–10.0 g/L resulted in a significant suppression of the pathogen growth (68.7–100%). On the seventh day of incubation, the observed antifungal effect of the minimal tested DMP concentration (1.0 g/L) did not exceed 32.4%. The DMP concentration causing 50% pathogen suppression was determined by a probit analysis to be equal to 2.5 g/L; this concentration was further used to determine the antifungal effect of DMP combined with fungicides. The calculated Pearson correlation coefficient ($r^2$) was 0.95, thus confirming a strong correlation between the studied parameters. The calculated *t*-criterion ($t_r = 5.51$) exceeded the threshold value of this criterion ($t_{crit} = 4.60$) determined by the table of *t* values at the significance level of 0.01, so the revealed correlation was statistically significant.

### 3.4. Effect of DMP on Conidial Germination in Botrytis cinerea

During germination, conidia form germ tubs, whose elongation is followed by further hypha branching. The results of the experiment are shown in Figure 5. Almost complete inhibition of conidial germination was observed for DMP concentrations of 5.0 and 10.0 g/L (Figure 5E,F). In the case of a DMP concentration of 2.5 g/L, germ tubes started to form at the end of the first 24 h of incubation (Figure 5D). DMP concentrations of 0.625 and 1.25 g/L (Figure 5B,C) were insufficient for inhibition of conidial germination of the pathogen; the length of fungal hyphae observed in these variants was comparable with that of the control variant (Figure 5A).

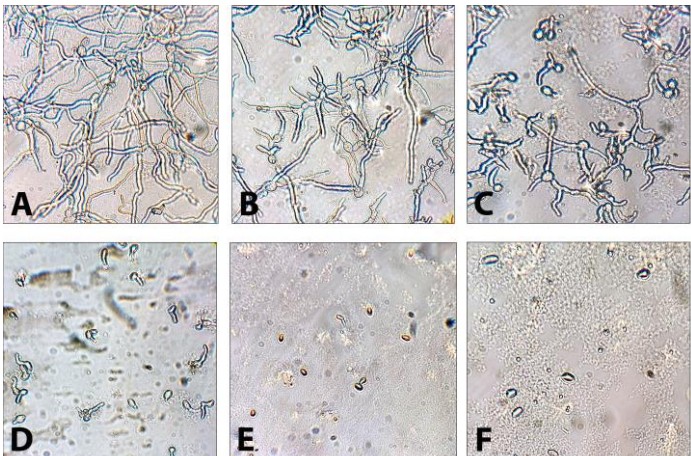

**Figure 5.** Dependence of the conidial growth inhibition in *Botrytis cinerea* on the DMP content in potato-glucose broth (g/L): (**A**) 0.0 (control); (**B**) 0.625; (**C**) 1.25; (**D**) 2.5; (**E**) 5.0; (**F**) 10.0 (100× magnification).

### 3.5. Evaluation of the Antifungal Activity of Fungicides (Difenoconazole, Tebuconazole, and Fludioxonyl) against B. cinerea

The study included the following fungicides: difenoconazole, tebuconazole (both at concentrations of 0.1, 1.0, and 10 mg/L), and fludioxonyl (0.01, 0.1, and 10 mg/L). The results of this experiment are presented in Figure 6.

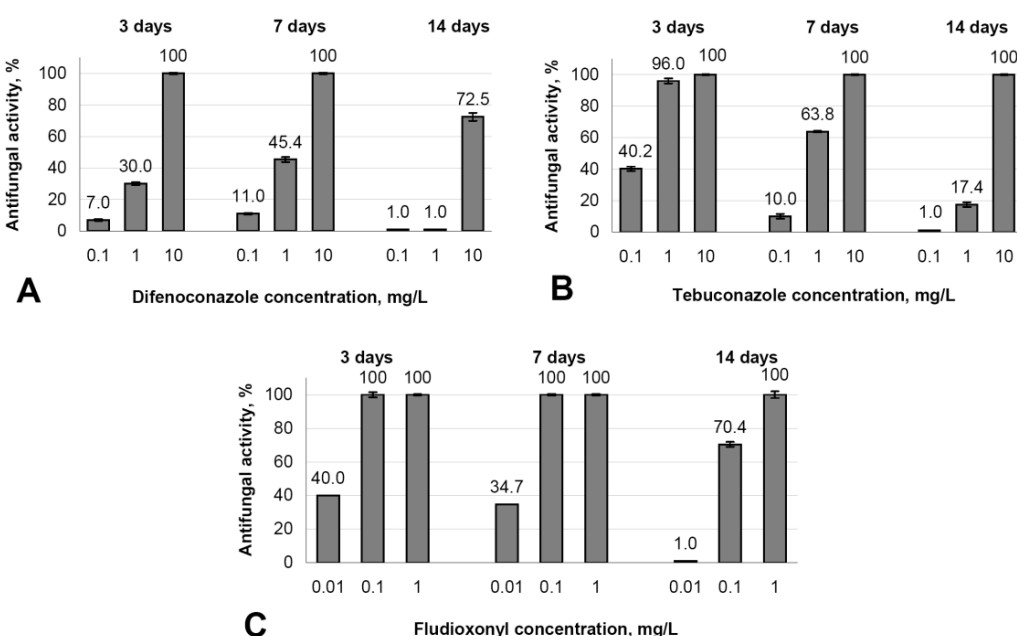

**Figure 6.** Antifungal activity (%) of (**A**) difenoconazole (0.1–10.0 mg/L), (**B**) tebuconazole (0.1–10.0 mg/L), and (**C**) fludioxonyl (0.01–1.0 mg/L) against *Botrytis cinerea*.

According to the obtained data, on the seventh day of incubation, the maximum (100%) growth inhibition of *B. cinerea* was observed for difenoconazole taken at a concentration of 10.0 mg/L (Figure 6A). On the fourteenth day of incubation, the pathogen spread almost over the whole Petri plate surface, and the antifungal activity of the fungicide was reduced to 72.5%. The fungicide concentration corresponding to 50% antifungal activity (effective concentration) was calculated by probit analysis and was equal to 1.0 mg/L. In the case of a minimal fungicide load (0.1 mg/L), the antifungal effect was almost absent.

In the case of tebuconazole (Figure 6B), a high inhibiting activity was observed within a concentration range of 1.0–10.0 mg/L; on the seventh day of incubation, it was 63.8% for 1.0 mg/L and 100% for 10.0 mg/L. On the fourteenth day of incubation, no pathogen growth was observed for the fungicide concentration of 10.0 mg/L. For the tebuconazole concentration of 0.1 mg/L, suppression of the pathogen growth was observed only on the third day of incubation; later, the pathogen grew over the whole Petri plate surface. The effective concentration of tebuconazole calculated by the probit analysis was 0.5 mg/L.

A high inhibiting activity of fludioxonyl was shown for the whole period of observation within the concentration range of 0.01–1.0 mg/L (Figure 6C). For example, the growth of *B. cinerea* still remained completely suppressed to the seventh day of incubation; to the end of the experiment, no pathogen growth was observed only for the variant with the maximal concentration (1.0 mg/L). The effective concentration calculated by the probit analysis was 0.04 mg/L.

### 3.6. Antifungal Activity of DMP Combined with Different Fungicides against B. cinerea

The DMP concentration used in this experiment was 2.5 g/L. Three fungicides, difenoconazole (D), tebuconazole (T), and fludioxonyl (F), were used at their effective concentrations (1.0, 0.5, and 0.04 mg/L, respectively). The results of the experiment are shown in Figures 7 and 8.

According to Figure 7, the studied combinations of DMP and fungicides effectively suppressed the growth of pathogen colonies for the first 7 days of the experiment. For example, DMP+T or DMP+F combinations completely suppressed the pathogen growth for the first 3 days of the experiment. Then the antifungal effect gradually decreased, reaching 86.5 and 84.6% on the seventh day and 28.5 and 37.4% at the fourteenth day for DMP+T and DMP+F, respectively.

In the case of the DMP+D combination, the maximum suppression of *B. cinerea* (85.6%) was observed on the seventh day of incubation, with further decreases. Note that no significant growth of *B. cinerea* was shown for this combination up to the end of the experiment, and the suppressing effect at the fourteenth day remained at the level of 51.3%.

A possible synergistic effect of the combined application of DMP and fungicides was evaluated using Limpel's formula (Formula (3)). A comparison of the calculated $E_e$ and $E_r$ values showed that the antifungal effect observed experimentally ($E_r$) on the seventh day of incubation exceeded the calculated $E_e$ values by 10% on average. $E_e$ values obtained on the third day of incubation for the DMP+T and DMP+F combinations exceeded the corresponding $E_r$ values by 23.7 and 26.8%, respectively. For the DMP+D variant, the $E_e$ and $E_r$ values were almost similar on the third day of the experiment, while the difference between these values on the fourteenth day of incubation was 36.1%.

The estimation of the significance of differences between samplings using the least significant difference method showed a significant difference between the antifungal activity of combined preparations and DMP or corresponding fungicides taken alone. A comparison of the antifungal activity levels of DMP or fungicides alone showed a lack of significant differences between these indices for both days 7 and 14 of the experiment.

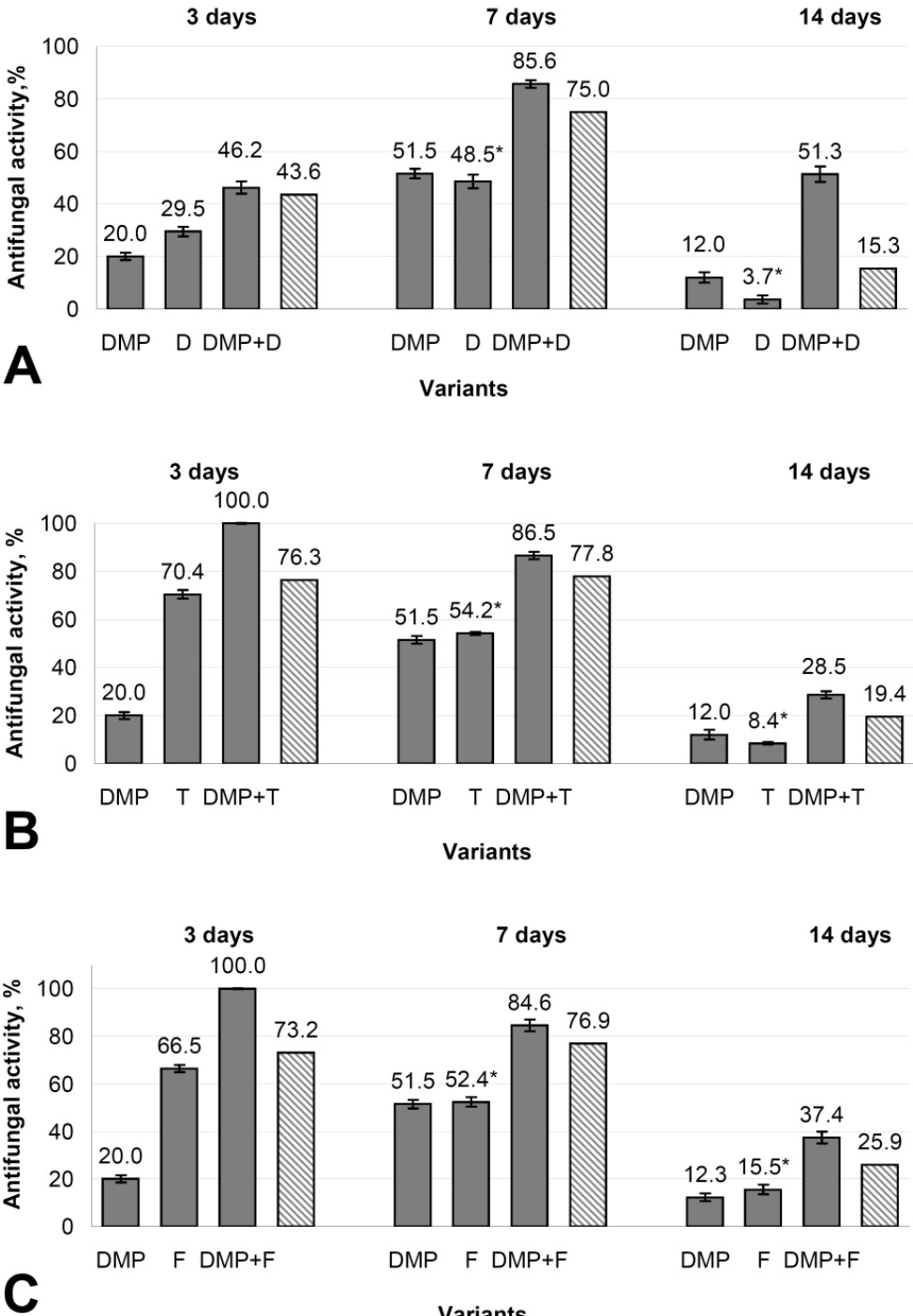

**Figure 7.** Antifungal activity of a dry mycelium biomass of *Penicillium chrysogenum* VKM F-4876D (DMP, 2.5 g/L) combined with (**A**) difenoconazole (D, 1.0 mg/L), (**B**) tebuconazole (T, 0.5 mg/L), and (**C**) fludioxonyl (F, 0.04 mg/L) against *Botrytis cinerea*. The hatched bar shows the expected antifungal activity ($E_e$) calculated by Limpel's formula. Values that were not significantly different from the values obtained for the DMP only are indicated by the asterisk.

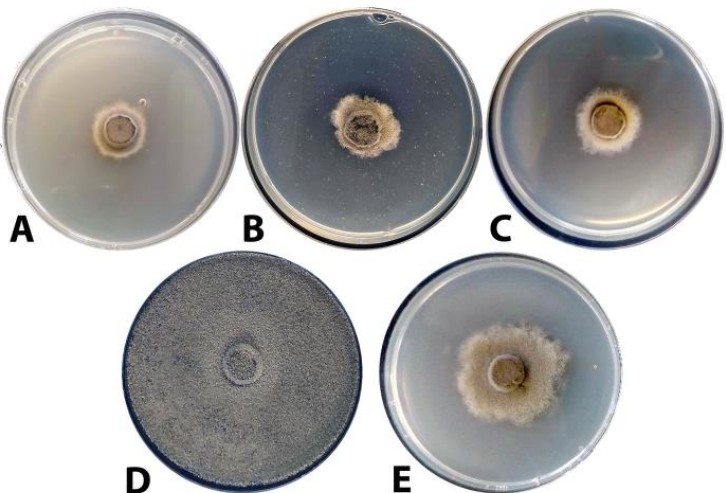

**Figure 8.** Colony growth of *B. cinerea* on the PDA medium supplemented with DMP (2.5 g/L) combined with the tested fungicides: (**A**) DMP + fludioxonil, 0.04 mg/L; (**B**) DMP + tebuconazole, 0.5 mg/L; (**C**) DMP + difenoconazole, 1.0 mg/L; (**D**) control; (**E**) DMP.

*3.7. Inhibition of Botrytis cinerea Growth on Grape Berries*

The DMP efficiency against *B. cinerea* on grape berries was evaluated on days 3, 7, and 12 of the experiment (Figure 9). According to the obtained results, no berries with minimal levels of infection (1–4 scores) were observed in the control on the seventh day of the experiment, whereas their fraction in the DMP-treated variant reached 80% on average (Figures 9 and 10). On day 14, a rapid development of grey mold was observed in the control variant (Figures 9C and 10C). Untreated grape berries were very soft and completely covered by *B. cinerea* mycelium. At the same time, the fraction of grape berries with a 9-score infection level in the DMP-treated variant was significantly lower and did not exceed 5% on average (Figure 10F). Moreover, the percentage of low (3–4 scores), medium (5–6 scores), and severe (7–8 scores) infections of DMP-treated berries was 40, 43.3, and 11.7%, respectively. Therefore, one can conclude that the treatment of grape berries with DMP (2.5 g/L) provides an efficient control on *B. cinerea* growth.

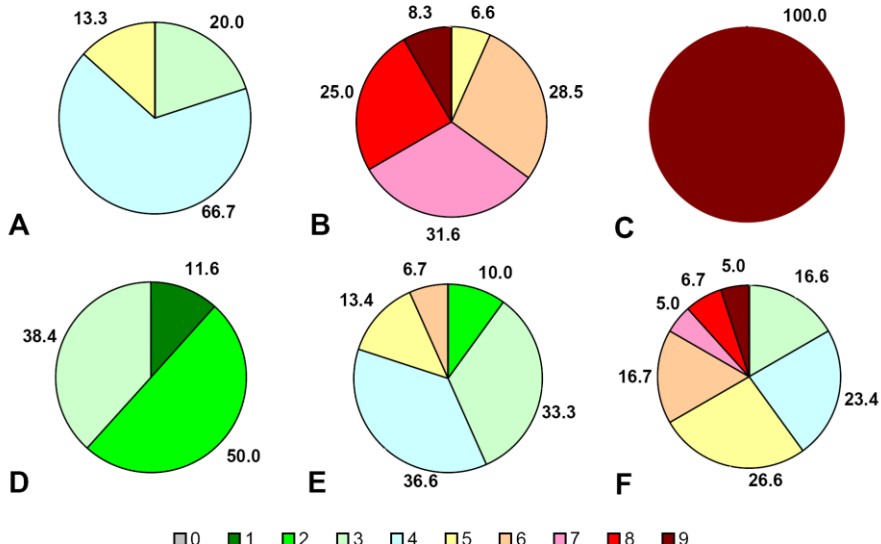

**Figure 9.** Level of affection of DMP-treated grape berries (cv. Husain) with *Botrytis cinerea* evaluated on the third (**A**,**D**), seventh (**B**,**E**), and twelfth (**C**,**F**) days of the experiment using a 0–10-score scale. (**A**–**C**), control; (**D**–**F**), DMP-treated berries.

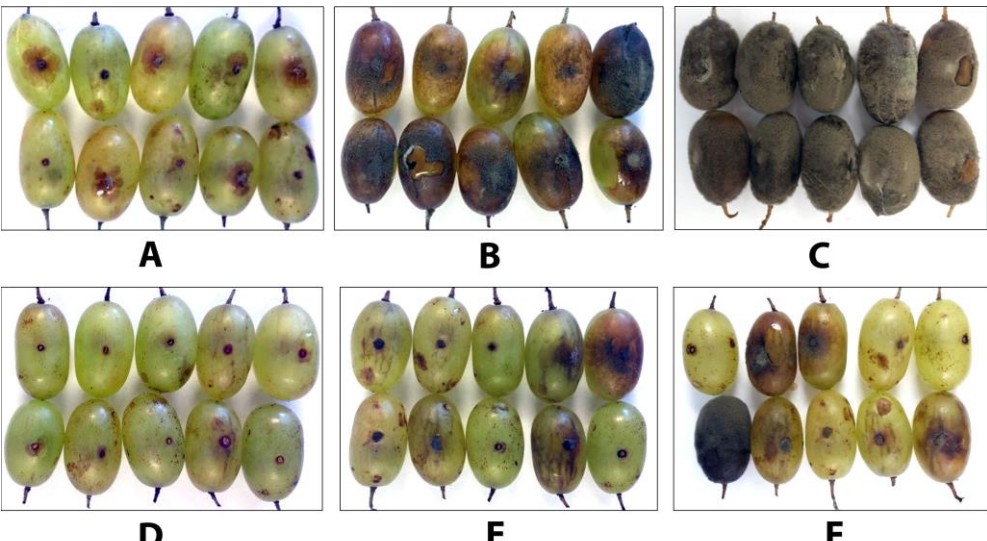

**Figure 10.** Dynamics of the infection of grape berries (cv. Husain) with *Botrytis cinerea* after the pre-treatment with DMP (2.5 g/L) determined on days 3 (**A**,**D**), 7 (**B**,**E**), and 14 (**C**,**F**) of the experiment. (**A**–**C**), control; (**D**–**F**), treated variant.

## 4. Discussion

Today, modern systems of protection of vegetable, fruit, and berry crops against gray mold are based on the use of chemical pesticides [39,40]. This fact is explained by both the peculiarities of the *B. cinerea* lifecycle and the insufficient efficiency of protection schemes based on biological control alone. Therefore, a complete rejection of chemical fungicides is rather inexpedient. Nevertheless, one should take into account that studies performed in different countries (China, France, Greece, and the USA) reported a high frequency of the development of *B. cinerea* resistance to the majority of chemical fungicides applied as mono- and combined preparations. The development of multiresistant strains of the pathogen poses a serious threat and significantly complicates its control by chemical protection only [41].

At the same time, the combined use of chemical and biological preparations can be considered an alternative method of grey mold control. On the one hand, these methods would enhance the fungicidal activity of modern preparations at low working concentrations; on the other hand, they would minimize the risk of the appearance of resistant pathogen strains.

The ability of *Penicillium* fungi to synthesize various biologically active compounds possessing fungicidal activity makes them promising for use in the development of biofungicide preparations [42,43]. We evaluated the antifungal activity of DMP, alone or combined with fungicides, in relation to the grey mold pathogen. The *P. chrysogenum* strain F-4856D used for DMP production was obtained from the *P. chrysogenum* VKM F-1310 strain using UV-induced multi-step mutagenesis coupled with a selection [31]. Earlier, we determined the DMP ability to suppress the growth of *Sclerotinia sclerotiorum* (Lib.) de Bary [44], some pathogenic *Fusarium* fungi (*Fusarium oxysporum*, *F. graminearum*, *F. avenaceum*, and *F. culmorum*) [31], and *Alternaria solani* [45] when used both individually and in combination with some fungicides.

In our previous study, we identified 6-demethylmevinolin, or mevastatin, as a component of the culture broth of F-4856D [44]. This compound possesses antifungal properties due to its ability to inhibit the biosynthesis of ergosterol, a basic cell wall component in fungi [46,47]. A microscopic study of *B. cinerea* mycelium directly contacting the sterile zone (Figure 3D,E) during a co-cultivation with F-4876D showed some structural changes of fungal hyphae (deformation and increased septation) and almost complete absence of conidia. Similar results were obtained for a co-cultivation of F-4876D with *S. sclerotiorum* [43]. The inhibiting effect of the fungal biomass components, including mevastatin, on conidial

germination of *B. cinerea* (Figure 3) agrees with other published data [48]. This fact can be very interesting for the further development of approaches to controlling this pathogen.

According to the obtained data, DMP addition to a nutrient medium at concentrations ranging from 7.5–10.0 g/L suppressed *B. cinerea* growth by 80.6–100% (Figure 2). Our calculations showed that the DMP concentration of 2.5 g/L provides a 50% suppression of *B. cinerea*. This concentration was further used for experiments with combinations of DMP and fungicides.

Tebuconazole and difenoconazole included in this study belong to the triazole group, the largest group of fungicides, whose effect is associated with the suppression of the biosynthesis of ergosterol, the main component of cell walls in fungi. Complex preparations based on these compounds are widely used to control grey mold in vegetables, fruits, and berries [39,40,49,50]. Fludioxonyl, developed due to the identification of bacterial toxin molecules produced by *Pseudomonas pyricinia*, represents a wide-range fungicide whose activity is based on the inhibition of glucose phosphorylation during cell respiration, which affects the functioning of fungal cell membranes and results in an intracellular accumulation of glycerin, whose excess causes the death of fungal cells. Fludioxonyl-based preparations are also widely used to control grey mold.

According to the existing data (see review [16]), the rate of development of *B. cinerea* resistance to already approved and used chemical fungicides significantly surpasses the process of developing new efficient fungicides against this pathogen. Therefore, alternative solutions intended to overcome the existing resistance and improve the antifungal effect of commercial fungicides at dosages providing a minimal effect are very relevant. Our study showed that combining difenoconazole, tebuconazole, and fludioxonyl with DMP at concentrations providing the level of their antifungal activity on the seventh day of the experiment to be only 48.5–54.2% when taken alone makes it possible to suppress the development of *B. cinerea* by 84.6–86.5%. Among the studied variants, the most effective one providing the maximum *B. cinerea* suppression in in vitro experiments was a combination of DMP (2.5 g/L) with difenoconazole (1.0 mg/L); even on the fourteenth day of the experiment, the antifungal effect of this combination was 51.3%, whereas the effect of DMP combinations with tebuconazole and fludioxonyl was reduced to 28.5 and 37.4, respectively. These results indicate a high potential for the use of DMP together with fungicides to control this disease.

The results of the experiment on grape berry protection from *B. cinerea* proved the fungicidal effect of DMP revealed in in vitro experiments. Nevertheless, since the results obtained in vitro sometimes cannot be successfully reproduced *in planta*, further confirmation of these results under field conditions should be performed.

## 5. Conclusions

In spite of the rather limited number of studies related to combining biological and chemical control of plant pathogens, the prospects of this line of investigations are indisputable. A high antifungal effect of DMP from *P. chrysogenum* VKM F-4876D combined with fungicides revealed in our study demonstrates the possibility of a real reduction of fungicide dosages required to control grey mold and other plant pathogenic microorganisms. This, in turn, will reduce the negative impact on the environment. In addition, since plant pathogens cannot develop resistance to biopreparations, the results of our study may open up the prospects for successive control of crop diseases without the need to increase the dosages of applied fungicides.

**Supplementary Materials:** The following supporting information can be downloaded at: https://www.mdpi.com/article/10.3390/agronomy13102602/s1, Figure S1: Nucleotide sequence of the fragment of a 28S RNA gene from *Botrytis cinerea* determined using the ITS5-LR5 primer system.

**Author Contributions:** Conceptualization, V.D. and E.G.; methodology, N.K. and A.S.; software, A.O.; validation, E.G., N.K. and V.Y.; formal analysis, V.D.; investigation, N.K., A.O., K.P. and A.H.; data curation, V.Y.; writing—original draft preparation, N.K. and A.H.; writing—review and editing,

V.Y. and E.G.; visualization, K.P. and A.S.; supervision, E.G.; project administration, V.D.; funding acquisition, V.D. All authors have read and agreed to the published version of the manuscript.

**Funding:** The study was supported by the Ministry of Science and Higher Education of the Russian Federation in accordance with agreement no. 075-15-2022-318 (20 April 2022) on providing a grant in the form of subsidies from the Federal budget of the Russian Federation. The grant was provided for state support for the creation and development of a world-class scientific center "Agrotechnologies for the Future".

**Institutional Review Board Statement:** Not applicable.

**Data Availability Statement:** The authors declare that the data supporting the findings of this study are available within the main text of the manuscript and Supplementary Materials. Raw data are available from the corresponding author upon reasonable request.

**Conflicts of Interest:** The authors declare no conflict of interest.

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
