# Peer review of "Inhibition of the Growth of Botrytis cinerea by Penicillium chrysogenum VKM F-4876D Combined with Fludioxonil-, Difenoconazole-, or Tebuconazole-Based Fungicides"

_agronomy, doi:10.3390/agronomy13102602_

Round 1
Reviewer 1 Report
The paper is well structured, the bibliography used covers the topic and the key message is well depicted. The experimental results are carefully demonstrated.
Some, revisions are necessary.
1. The aim of the study should be rewritten.
2. Evaluation of antifungal activity for the synthetic fungicides is missing from “Materials and Methods” Should be added. Preparation of testing solutions, bioassays , etc
3. For EC50 estimation at least 4 different doses (concentrations ) should be used. In this study only 3 doses are used for chemicals, therefore it is suggested not to present the results as EC50 but only as % of antifungal activity.
4. Figure 6 is not correct.
5. Replace the “registered” with found or shown or estimated (highlighted blue)
6. correct some editing ( highlighted blue)
To sum up, I think the paper can be accepted after the above suggested revisions.
All these are highlighted blue on the text

none
Author Response
The paper is well structured, the bibliography used covers the topic and the key message is well depicted. The experimental results are carefully demonstrated.
We thank the reviewer for a positive evaluation of our study and appreciate his/her valuable comments.
Some revisions are necessary.
- The aim of the study should be rewritten.
Done.
- Evaluation of antifungal activity for the synthetic fungicides is missing from “Materials and Methods” Should be added. Preparation of testing solutions, bioassays , etc.
Additional information was added (see subsections 2.7, 2.8, 2.9).
- For EC50 estimation at least 4 different doses (concentrations ) should be used. In this study only 3 doses are used for chemicals, therefore it is suggested not to present the results as EC50 but only as % of antifungal activity.
The corresponding corrections were made. EC50 was replaced with “effective concentration corresponding to the 50% antifungal activity”.
- Figure 6 is not correct.
Yes, several values were incorrectly reflected on the graph. The problem was corrected.
- Replace the “registered” with found or shown or estimated (highlighted blue)
Done.
- correct some editing ( highlighted blue)
Unfortunately, the reviewer did not attach a file with blue highlights in the text.
To sum up, I think the paper can be accepted after the above suggested revisions.
All these are highlighted blue on the text
Reviewer 2 Report
Dear Authors
I find it an interesting article, well written. Thank you very much.
Did the authors investigate the effect of using low doses of antifungals on fungal strains for a longer period of time?
Is it possible to replicate the trial but using DMP at higher doses, but without antifungals?
What do the authors think about using low doses of antifungals? wouldn't this be counterproductive in the sense that at low doses, resistance to antifungals would be enhanced?
kind regards

Author Response
Dear Authors
I find it an interesting article, well written. Thank you very much.
Thank you for good appreciation of our work and for interesting questions.
Did the authors investigate the effect of using low doses of antifungals on fungal strains for a longer period of time?
No, we did not do such investigations for a long-time period in relation to our pathogen-fungicide systems. Nevertheless, such study will probably be performed in the future to check the possibility to obtain fungicide-resistant forms of plant pathogens and to study then the efficiency of the developed biopreparation in the overcoming of such induced resistance.
Is it possible to replicate the trial but using DMP at higher doses, but without antifungals?
Our study showed (Fig. 4) that in the case of a DMP concentration of 10.0 g/L in PDA, a complete suppression of the B. cinerea growth was observed for 7 days of the experiment. Therefore, use of DMP alone in high concentrations to suppress the pathogen development is possible, but is rather inexpedient for economical reasons. Moreover, use of DMP in high concentrations may cause some technical problems for the field treatments, since the agricultural machinery used for crop treatments is usually designed to work with water solutions.
What do the authors think about using low doses of antifungals? wouldn't this be counterproductive in the sense that at low doses, resistance to antifungals would be enhanced?
The method of plant protection developed in our studies is based on the chemosensitization principle used in medicine. This principle is based on the increased sensitivity of a pathogen to low doses of fungicide preparations, which, taken alone, usually do not provide any suppressive effect, but being combined with chemosensitizers, strongly suppress the fungal development. Chemosensitizing compounds attack biochemical or structural targets differing from those of fungicides and, therefore, do not promote selection of resistant forms. In our study, DMP is considered not only as a biopreparation possessing a high antifungal activity, but also as a chemosensitizer intended to increase B. cinerea sensitivity to chemical fungicides.
Thus, we suppose such method of protection may be considered as an efficient tool to control resistant strains of plant pathogenic fungi without an increase in the dosages of chemical fungicides.
Reviewer 3 Report
The author need little improvement in introduction , they should add article related to biocontrol they are using and if the biocontrol has been tested for pesticide resistance
Line 67 – give some example what can be helpful for example you can look into Srivastava et al 2021 (https://doi.org/10.3390/plants10020210) for reference
Line 81 – please give reference
Line 92 – this is not very well known please either give reference or remove this line
Line 95 – please give more details of the line P. chrysogenum VKM F-4876D , why it was chosen and where it was isolated from , also refer some paper if any relevant , Is this line has been characterized before ?
Line 129 – please elaborate more
Line 134 please write full name of organism. P. chrysogenum F-4876D be consistent with the name through out the manuscript
Line 144 – please mention pressure as well (15 psi)
Line 175 – fungi does not grow in perfect circle how was the diameters were taken at that time please mention that as well. If possible make a illustration of this method
Line 180 – please define size of Petri plate used?
Line 280 - Figure -3 define day after it
Line 320 – figure 5 – please include scale with the photographs. pictures are blue because the microscope was not properly focused please provide better pictures
line 410 - please also discuss the possibility of acquiring multiple resistance by fungi , one of the reason multiple drugs are not used in large numbers
English was good and grammar was also correct
Author Response
The author need little improvement in introduction, they should add article related to biocontrol they are using and if the biocontrol has been tested for pesticide resistance
Authors thank the reviewer for his/her valuable comments. We slightly enlarged introduction and discussion and added some referemces.
Line 67 – give some example what can be helpful for example you can look into Srivastava et al 2021 (https://doi.org/10.3390/plants10020210) for reference
We added some information and references.
Line 81 – please give reference
These data are from Ref. 16. Done.
Line 92 – this is not very well known please either give reference or remove this line
The reference was added.
Line 95 – please give more details of the line P. chrysogenum VKM F-4876D , why it was chosen and where it was isolated from , also refer some paper if any relevant , Is this line has been characterized before ?
This strain was obtained at our laboratory by a multi-step UV mutagenesis. The detailed information about it development and characterization can be found here (Ref. 31):
Karpova N.V., Yaderets V.V., Glagoleva E.V., Petrova K.S., Ovchinnikov A.I., Dzhavakhiya V.V. Antifungal activity of the dry biomass of Penicillium chrysogenum F-24-28 and its application in combination with azoxystrobin for efficient crop protection. AGRICULTURE 2021, 11(10), 935, doi: 10.3390/agriculture11100935.
The strain F-24-28 was deposited to the All-Russian Collection of Microorganisms as VKM F-4876D. Some our later publications described the efficiency of this strain combined with some fungicides against various plant pathogens (refs. 44 and 45 mentioned in the Discussion section).
We added the reference to the first mentioning of this strain in the text.
Line 129 – please elaborate more
Done.
Line 134 please write full name of organism. P. chrysogenum F-4876D be consistent with the name through out the manuscript
Done.
Line 144 – please mention pressure as well (15 psi)
Done.
Line 175 – fungi does not grow in perfect circle how was the diameters were taken at that time please mention that as well. If possible make an illustration of this method
This method is quite well illustrated in Fig. 3. First, the growth inhibition zone appearing during a co-cultivation of P. chrysogenum and B. cinerea is clearly visible. Second, in this study two parameters were measured: d1 is the distance between the centers of agar disks with B. cinerea and P. chrysogenum and d2 is the distance between the center of B. cinerea agar disk and the edge of its growth measured along the d1 line (see the ends of the subsection 2.6).
In relation to the experiments described in 2.9, the colony diameter is determined by the measurement of the maximum and minimum diameter of a colony and calculation of their mean. The corresponding information was added to 2.9.
Line 180 – please define size of Petri plate used?
We added information about the size of Petri plates used in our study.
Line 280 - Figure -3 define day after it
The data of observations is shown in Fig. 3 - this is the 7th days of co-cultivation.
Line 320 – figure 5 – please include scale with the photographs. pictures are blue because the microscope was not properly focused please provide better pictures
The scale size was identified. Unfortunately, we do not have other pictures, but even now they quite clearly illustrate the studied effect.
line 410 - please also discuss the possibility of acquiring multiple resistance by fungi , one of the reason multiple drugs are not used in large numbers
We added some information and reference concerning this issue into the Discussion section.